# Effects of Dietary Protein Level on the Gut Microbiome and Nutrient Metabolism in Tilapia (*Oreochromis niloticus*)

**DOI:** 10.3390/ani11041024

**Published:** 2021-04-05

**Authors:** Changgeng Yang, Ming Jiang, Xin Lu, Hua Wen

**Affiliations:** 1Life Science & Technology School, Lingnan Normal University, Zhanjiang 524048, China; yangcg910@163.com; 2Fish Nutrition and Feed Division, Yangtze River Fisheries Research Institute, Chinese Academy of Fishery Sciences, Wuhan 430223, China; luxing@yfi.ac.cn (X.L.); wenhua.hb@163.com (H.W.)

**Keywords:** tilapia, dietary protein, gut microbiome, nutrient metabolism

## Abstract

**Simple Summary:**

Dietary protein is an important factor affecting aquaculture. In this study, the homeostasis of the gut microbiome and metabolic profile of the liver and serum of tilapia were analyzed, comparing those fed with different diets to evaluate the effect of diet on protein levels. As a result, there was no significant difference found in the diversity and richness of the gut microbiome but had differences in the microbial composition of the gut among different groups. As for the liver metabolome of the tilapia, the glucose content increased along with increased protein levels. As for serum metabolome, the levels of tyrosine, guanosine, and inosine were significantly different. In summary, diets with different protein levels can affect the composition of gut microbiota and glycolysis and amino acid metabolism in tilapia. These results may also help to improve the conditions of tilapia cultivation.

**Abstract:**

Dietary protein is one of the most important nutritional factors in aquaculture. The aim of this study was to examine the effects of dietary protein levels on the gut microbiome and the liver and serum levels of metabolites in tilapia. Tilapia were fed a diet with a low (20%), moderate (30%), or high (40%) content of crude protein, and the homeostasis of the gut microbiome and metabolic profile of the liver and serum were analyzed. The results showed no significant differences in the diversity and richness of the gut microbiome among the groups; however, there were differences in the microbial composition of the gut. The metabolome analysis of liver samples revealed a difference in the glucose level among the groups, with the highest glucose level in fish fed a high protein diet. In addition, there were significant differences in the levels of tyrosine, guanosine, and inosine among the metabolome analysis of serum samples of these groups. In summary, diets with different protein levels could affect the composition of gut microbiota and the dynamic balance of microbial communities. Dietary protein content can also affect glycolysis and amino acid metabolism in tilapia.

## 1. Introduction

Fish require protein in their diets for growth and development [1]. Different dietary protein levels can affect the growth, gut microbial composition, nutrient metabolism, and various physiological reactions in fish. In cases of low dietary protein, the activity of various digestive enzymes is decreased, and growth is retarded, ultimately affecting lifespans [2,3]. In cases of high dietary protein level, growth is also retarded; however, instead, the digestion and absorption of nutrients is suboptimal [4]. At the same time, fish can use stored proteins for energy consumption, which increases the excretion of ammonia nitrogen and promotes pollution, which is not conducive to the health of fish and the sustainable development of ecological environments [4,5,6].

Tilapia belong to the order Perciformes, family Cichlidae, species *Oreochromis niloticus*. Tilapia is widely cultivated in many countries, and it is one of the main farmed fish in China. Therefore they require high-quality feed that promotes growth and development. In recent years, researchers and farmers are constantly striving to improve the protein utilization and feed efficiency of diets in order to promote healthy farmed tilapia [7,8,9]. For example, the research on protein requirement of tilapia at different stages, the analysis of the utilization of different protein sources, analysis of the effect of different protein sources and different protein level on the digestive system, the development of various additives to improve the efficiency of protein utilization, and the in-depth analysis of tilapia response to feed protein at the molecular level [10,11,12,13,14].

Recently, many new methods have been used in nutrition research. For instance, high-throughput sequencing could help characterize the gut microbiome of fish fed with different diets; transcriptome sequencing and metabolome analysis may also be used in exploring the transcriptomic and metabolic response of fish to different diets. These methods provide us new approaches in tilapia nutrition studies. As for investigations of tilapia diets, Candis et al. used 454 pyrosequencing to characterize the gut microbiome of fish fed diets containing prebiotics [15]. Zheng et al. used high-throughput RNA sequencing technology to explore the global transcriptomic response of hepatic mRNA of tilapia fed with diets containing Resveratrol [16]. ^1^H NMR-based and GC-MS-based metabolomics approaches were applied to investigate the metabolite variations in Nile tilapia fed with various dietary supplementation [17,18]. There is currently some research on the optimal dietary protein level for the growth and development of tilapia, which has been reported to be 30–35% [2,9,19]. On this basis, we aimed to examine the effects of dietary protein level on the gut microbiome of tilapia using high-throughput sequencing, as well as the effects of dietary protein level on the nutrient metabolism using nuclear magnetic resonance studies. These results will increase our understanding of the effects of dietary protein on the physiological processes in tilapia from the perspective of nutritional metabolomics, which may facilitate the discovery of new biometabolic markers. These results may also help to improve the conditions of tilapia cultivation.

## 2. Materials and Methods

### 2.1. Preparation and Management of Tilapia

Tilapia (initial body weight, 38.75 ± 0.61 g) were collected from a hot spring farm in Xianning (Hubei Province, China) and transported to the Yangtze River Fisheries Research Institute for indoor breeding in a recirculating aquaculture system. Tilapia were randomly divided into three groups, namely the low dietary protein group (LP group), moderate dietary protein group (MP group), and high dietary protein group (HP group). Each group had three replicates. Each replicate with 25 tilapia, respectively, was raised in independent tanks. The low, moderate, and high dietary protein groups were supplemented with 20%, 30%, and 40% of crude protein, respectively. Each group was raised in a separate tank for 8 wks. Fish were fed three times daily (at 08:30, 12:30, and 16:40) until apparent satiation was reached within 30 min for 8 weeks [20]. The tank temperature was 28 °C ± 1 °C, the pH was 7.1–7.4, the dissolved oxygen level was >5.0 mg/L, and the light was natural. The details of each diet are provided in Table 1.

The care, handling, and sampling of fish were performed following animal care protocols approved by the Animal welfare committee of Yangtze River Fisheries Research Institute, Chinese Academy of Fishery Sciences.

### 2.2. Sample Collection

After feeding for 8 wks, tilapias were fasted for 24 h. The final body weight of each tilapia was recorded, and the weight gain rate (WGR) and specific growth rate (SGR) was calculated as follows
weight gain rate (WGR) = 100 × (FBM − IBM)/IBM
specific growth rate (SGR) = 100 × (lnFBM − lnIBM)/t
where FBM is the final body weight of each tilapia in a group after t days (g), IBM is the initial body weight of each tilapia in a group (g), and t is the number of rearing days.

Six fish from each group (2 fish from every tank) were randomly selected and anesthetized with 75 mg/L MS-222. The midgut samples were cut and directly collected on the ice for high-throughput analysis of the gut microbiome. Blood was collected through the tail vein, and the liver was dissected and separated from these fish. The livers were rapidly frozen in liquid nitrogen. The blood samples were placed in a refrigerator at 4 °C for 30 min and then centrifuged at 3000 r/min for 5min. The upper serum was taken and frozen at −80 °C.

### 2.3. Extraction of Total DNA from Intestinal Samples and Polymerase Chain Reaction (PCR) Amplification

DNA was extracted from intestinal samples using the QIAamp DNA Stool Mini Kit (Qiagen, Germantown, MD, USA). Q5 High-Fidelity DNA Polymerase (NEB, Ipswich, MA, USA) was used for PCR amplification. The primer sequences used were for the V3 (actcctacgggaggcagca) and V4 (ggactachvgggtwtctaat) regions. The PCR amplification conditions were as follows: initial denaturation at 94 °C for 5 min; followed by 30 cycles of denaturation at 94 °C for 1 min, annealing at 55 °C for 1 min, and extension at 72 °C for 1.5 min; and a final extension at 72 °C for 10 min. The PCR products were visualized on 2% agarose gels, and the target DNA fragments were recovered using the AxyPrep DNA Gel Extraction Kit (Axygen, San Francisco, CA, USA). Two-terminal 2 × 300 bp reads of the recovered DNA were sequenced using the Illumina MiSeq System.

### 2.4. Processing of Liver and Blood Samples for Nuclear Magnetic Resonance (NMR) Data

Liver samples: Fifty micrograms of liver were weighed, combined with 1 mL of purified water, and homogenized (Sonics VX-130, Newtown, CT, USA) eight times using a program of 4 s of homogenization and 3 s of cooling on ice. The samples were centrifuged at 13,000 rpm for 15 min at 4 °C, followed by re-centrifugation through an ultrafiltration membrane (Millipore Amicon ULTRA3 ku, Billerica, MA, USA) at 13,000 rpm for 45 min at 4 °C. Thereafter, 450 µL of the supernatant was combined with sodium trimethylsilylpropanesulfonate (DSS), vortexed for 10 s, and centrifuged at 13,000 rpm for 2 min at 4 °C. The samples were used for further analysis.

Serum samples: The samples were centrifuged at 13,000 rpm for 2 min at 4 °C, followed by re-centrifugation through an ultrafiltration membrane at 13,000 rpm for 65 min at 4 °C. Thereafter, 450 µL of the supernatant was combined with DSS, vortexed for 10 s, and centrifuged at 13,000 rpm for 2 min at 4 °C. The samples were used for further analysis.

Spectral data were collected using a Bruker AV III 600 MHz spectrometer equipped with an inverse cryoprobe. The parameters are provided in Table 2.

### 2.5. Statistical Analysis

For analysis of the gut microbiome, raw sequences were filtered by the Quantitative Insights Into Microbial Ecology (QIIME) Tool. The following databases were used as OTU taxonomic status identification databases: Greengenes (Release 13.8, http://greengenes.secondgenome.com/ (accessed on 26 June 2018)), Silva (Release 115, http://www.arb-silva.de (accessed on 26 June 2018)), UNITE (Release 5.0, https://unite.ut.ee/ (accessed on 26 June 2018)). According to Bokulich [21], any OTU with an abundance of less than 0.001% of the total sequencing of the whole sample was removed. Sequences were assigned to operational taxonomic units (OTUs) at 97% similarity. To estimate the alpha diversity (richness and diversity within the samples), four metrics were calculated, namely the ACE index, Chao1 index, Shannon index, and Simpson index. At phylum and genus levels, the number of microbial communities in each sample was determined. The linear discriminant analysis (LDA) effect size (LEfSe) method (http://huttenhower.sph.harvard.edu/galaxy/ (accessed on 26 June 2018)) was used to compare the abundance of all detected bacterial taxa among three groups’ fish. ANOVA was used to analyze the significance of the differences between the different groups, and *p* < 0.05 was considered a significant difference.

For analysis of metabolites, free induction decay signals were zero-filled, and Fourier transformed using the processing module within Chenomx NMR Suite 8.1 Software (Chenomx Inc., Edmonton, AB, Canada). Data were phased and baseline-corrected using the same module. Spectral data were referenced to the internal standard, and DSS was compared against the Chenomx Compound Library. Forty metabolites from 30 spectra were identified and quantified. Data were exported to Microsoft Excel, normalized by weight, and used in multivariable analysis. PLS-DA was performed using the pls package [22]. Plots were generated using the ggplot2 package [23]. ANOVA was used to analyze differences in the VIP scores of metabolites. All data are presented as mean ± standard deviation (SD) unless indicated otherwise.

## 3. Results

### 3.1. Growth Performance of Tilapia Fed Different Diets Enriched with protein

The growth performance of tilapia fed diets with different proportions of crude protein for 8 wks is provided in Table 3. We found that the WGR and the SGR were significantly higher in MP and HP groups than those in the LP group (*p* < 0.05), whereas there were no significant differences in these parameters between MP and HP groups (*p* > 0.05).

### 3.2. Gut Microbiome Analysis

#### 3.2.1. Analysis of Gut Microbiome Sequencing Results

As shown in Table 4, a total of 1,100,104 sequences were obtained from all samples, and the maximum number of sample sequences was 71,516, and the minimum was 52,611. A total of 3750 OTUs were generated for the three dietary protein level groups. As shown in Figure 1, there were 2280 (68.80%) common OTUs among the three groups. Furthermore, there were 2709 (75.50%) common OTUs between the HP and MP groups, 2491 (67.95%) common OTUs between the LP and MP groups, and 2603 (68.28%) common OTUs between the HP and LP groups. All raw sequences are available in the NCBI Sequence Read Archive under BioProject ID: PRJNA700685.

#### 3.2.2. Alpha Diversity Analysis of The Gut Microbiome

As shown in Table 5, there were no significant differences in the diversity indexes (Simpson index and Shannon index) and richness indexes (Chao1 index and ACE index) among the three dietary protein groups (*p* >0.05).

#### 3.2.3. Taxonomic Composition of the Gut Microbiome

On the level of phylum, in all groups, Proteobacteria, Fusobacteria and Firmicutes were the dominant phyla, comprising 80% of the total phyla in the intestine, as shown in Figure 2. The proportion of Proteobacteria was similar across fish in the MP group, whereas that of Fusobacteria was higher in the LP and HP groups but lower in the remaining group. The proportions of other phyla, such as Actinobacteria and Bacteroidetes, varied across different individuals and accounted for smaller proportions.

At the genus level, in all groups, the proportion of *Cetobacterium, Pseudomonas, Unclassified Clostridiaceae, Unclassified Rhizobiales, Clavibacter* were the top five, accounting for about 60% of the total. Furthermore, the proportion of *Cetobacterium* was higher in the LP and HP groups but lower in the remaining group. Other genera, such as *Pseudomonas*, the order Rhizobiales, and the family Clostridiaceae, accounted for certain proportions (Over 6% on average) in the three groups, and there were differences among different individuals, as shown in Figure 3.

#### 3.2.4. Analysis of the Linear Discriminant Analysis (LDA) Effect Size (LEfSe)

Through LEfSe analysis, the abundance of Clostridium was found to be significantly higher in the HP group, the abundance of *Enterovibrio* and *Grimontia* were significantly higher in LP (*p* < 0.05), at the genus level, and there was no significant difference at the phylum level (*p* > 0.05), as shown in Figure 4.

### 3.3. ^1^H NMR Metabolomics Analysis

#### 3.3.1. ^1^H NMR Metabolite Profiles In liver and Serum Samples

As shown in Figure 5, a total of 45 metabolites were identified in the liver samples, namely 22 amino acids or amino acid derivatives, seven nucleic acids, six organic acids, four amines, and ammonia compounds, three sugars, and three other compounds based on the data retrieved from the Chenomx database. In addition, a total of 47 metabolites were identified in the sera, namely 11 organic acids, 20 amino acids or amino acid derivatives, three nucleic acids, three sugars, two alcohols, four amines and ammonium compounds and two other compounds.

#### 3.3.2. Partial Least Squares Discriminant Analysis (PLS-DA) of Liver and Serum Samples

In the PLS-DA score plots of the metabolite in the liver (Figure 6a), there were few differences in metabolites in each LP, MP, and HP groups, respectively. However, the LP, MP, and HP groups showed a trend of separation, and the metabolites among the three groups were different. From the PLS-DA loading plots of the metabolite in the liver (Figure 6b), Most of the data points were clustered near the origin, and only a small number of data points, such as glucose and maltose, were scattered. The contribution rate of metabolites such as glucose and maltose to the inter-group differentiation of samples is large. The PLS-DA permutation test of the liver data indicates that the classification by the PLS-DA model was good (*p* = 0.003) (Figure 7).

From the PLS-DA score plots of the metabolite in the serum (Figure 8a), there was no difference in metabolites between each of the LP, MP, and HP groups, respectively. However, the LP, MP, and HP groups showed a trend of separation, and the metabolites among the three groups were very different. From the PLS-DA loading plots of the metabolite in the liver (Figure 8b). Most of the data were clustered near the original place, and a few of the data points were relatively discrete, indicating differences in metabolites between the three groups. The PLS-DA permutation test of serum data indicates that the classification by the PLS-DA model was good (*p* = 0.003) (Figure 9).

#### 3.3.3. Variable Importance in Projection (VIP) Analysis of Liver and Serum Metabolites

As shown in Figure 10, VIP analysis of liver samples from the three dietary protein level groups identified 15 key metabolites, with glucose and maltose showing differences in the VIP score among the groups (VIP > 1). The results of analysis of variance (ANOVA) performed on metabolites with VIP scores greater than 1 showed a significant difference for glucose (*p* < 0.05), but not for maltose (*p* > 0.05), among the groups. The remaining metabolites were amino acids and amino acid derivatives; however, no significant differences in the VIP score were observed among the groups (VIP < 1).

As shown in Figure 11, VIP analysis of serum samples from the three dietary protein level groups identified lactic acid, glucose, guanosine, tyrosine, alanine, inosine, phenylalanine, leucine, valine, and isoleucine as the key metabolites with VIP scores greater than 1. However, the results of ANOVA only showed significant differences for tyrosine, guanosine, and inosine among the groups (*p* < 0.05).

## 4. Discussion

In this study, tilapia were fed diets with different proportions of crude protein. After 8 wks of feeding, the growth rate was determined. We found that a diet containing 30% protein could satisfy the growth of tilapia at 28 °C. This is similar to other studies [2,9].

Fish gut microbiota play important roles in the breakdown and absorption of nutrients, in gut immunity, as well as in host health [24,25,26,27]. The colonization, establishment, composition, and diversity of intestinal microflora in fish is a complex process, which is a comprehensive reflection of the influencing factors such as aquaculture water, feed, environment microorganisms, and their development stage [28,29,30,31]. In fact, many studies on fish nutrition have found that diet is the main factor affecting the composition and metabolism of gut microflora. Meanwhile, the diversity and species composition of intestinal microflora is different for different fish and under different environmental conditions, which are more complex than we think. For example, in the research on the replacement of fish meal with plant protein sources, it was found that there was no significant change in the diversity of gut microbiome in juvenile olive flounder and juvenile hybrid grouper [32,33]. However, some studies in Atlantic salmon fed with different protein sources diet showed significant changes in bacterial communities in the gut [34].

Some studies have reported changes in the gut microbiome after altering the diet protein level [14,35]. Ideally, an optimal diet can maintain a healthy intestinal environment, promote nutrient decomposition, and facilitate rapid growth [36]. Here, high-throughput sequencing was used to analyze the gut microbiome of tilapia fed diets with different proportions of protein. There were no significant differences in species richness and diversity among the three dietary protein level groups. This result is slightly different from those of Zhu et al. research [14]. In their study, the intestinal microbial diversity of tilapia in the low protein diet group (25% protein diet) was significantly decreased compared with the control group (35% protein diet). This difference may be due to the experimental objects, the experimental environment, fasting time before sampling, and the feed formula was slightly different from ours. For example, the crude lipid in our diet was significantly higher than that of the Zhu et al. experiment. At the same time, the initial body weight of our experimental fish was 38.75 ± 0.61 g, while the initial weight of fish in the Zhu et al. experiment was about 0.8 g. The intestine of juvenile fish in the Zhu et al. experiment is in the initial development stage, which may be more easily affected by external factors, resulting in diversity differences. Furthermore, fasting can have a high impact on the intestinal microbiome; fish were fasted for 24 h in our research and overnight in the Zhu et al. experiment, which might be the reason for the difference in the results.

Although in our research, there was no difference in intestinal microbial diversity, it was found that at the genera level, the proportion of the same microbes in different groups of tilapia and intestinal microflora composition was slightly different. The predominant microorganisms in the gut of fish are usually Firmicutes, Proteobacteria, Bacteroidetes, and Fusobacteria, although various factors, such as the water temperature, feeding conditions, and other dietary factors, can affect the intestinal environment [36,37,38]. In tilapia, the predominant microorganisms in the gut are Proteobacteria, Firmicutes, Actinobacteria, and Fusobacteria, consistent with the results of our study, which showed high proportions of Fusobacteria, Proteobacteria, and Firmicutes in the three dietary protein level groups [39,40,41]. However, the proportion of Fusobacteria in the MP group was smaller than that in the other two groups. Fusobacteria is a small group of Gram-negative bacteria. It plays a role in promoting nutrition metabolism, digestion, and absorption, and some species of Fusobacteria can also lead to some diseases [42,43]. For example, *Fusobacterium nucleatum* can cause opportunistic infections [44]. Proteobacteria is the largest phyla of bacteria, distributed in a variety of environments, including photosynthetic, inorganic species, and many pathogens. The change in the abundance of Proteobacteria in the intestine is an important sign of the imbalance of intestinal flora [45]. The relative abundance between Firmicutes and Bacteroidetes is associated with obesity, and together, they promote the host efficiently absorb the energy in food [46]. In this study, the abundance of Firmicutes in most groups was higher than that of Bacteroidetes, which might be related to energy metabolism. The specific reasons need further verification. At the genus level, *Clostridium* (Firmicutes; Clostridia; Clostridiales; Clostridiaceae) was one of the most common genera of endogenous flora in the gut of freshwater fish [47]. *Clostridium* is one of the main protein-degrading bacteria [48], and it is also the main cellulose-degrading bacteria in the intestine environment [49]. We found that the abundance of *Clostridium* is the highest in the HP group. The reason may be that protein content as well as the micro-cellulose was both the highest in the HP group. *Enterovibrio* and *Grimontia* (Proteobacteria; Gammaproteobacteria; Vibrionaceae) are pathogenic bacteria, and changes in their abundance could cause host intestinal microbiology imbalance. In our research, the relative abundance of *Enterovibrio* and *Grimontia* in the LP group was significantly higher than that in the other two groups. This may be due to the lower diet protein content in the LP group, which cannot meet the needs of tilapia; this might result in impacts to disease resistance. On the contrary, the abundance of *Enterovibrio* was very low, which may have a limited effect on the gut of tilapia.

As a highly sensitive and highly accurate approach, ^1^H metabolomics has gained momentum in the field of aquaculture. For example, Wei et al. studied the effects of protein hydrolysates on liver and muscle function in juvenile *Scophthalmus maximus* using this approach and identified different compounds related to amino acid metabolism and glycolysis [50]. On the other hand, Panita et al. observed that a diet high in fat and carbohydrate can cause metabolic disorders in *Megalobrama amblycephala* [51].

The liver is an important metabolic organ, which plays an important role in the synthesis and decomposition of proteins, fats, and carbohydrates, the mutual transformation of the three, and the balance of body energy. Secondly, in biological organisms, blood is an important carrier to transport oxygen, nutrients, cell metabolites to various organs and cells. In this study, using ^1^H-NMR metabolomics, we identified 45 and 47 metabolites in the liver and blood of tilapia fed different diets, respectively.

In our results, we compared liver tissue metabolites of three groups of fish fed with different protein levels of diets; a significant difference in glucose content was found in each group. As protein levels increased, glucose content increased. The difference metabolite was found to be glucose, which indicated that the protein content in the diet had a significant effect on the carbohydrate metabolism of tilapia livers. Because the carbohydrate in the diet was not used as effectively as the protein in the diet, so glycolysis and gluconeogenesis in the carbohydrate metabolism of the fish are especially important [52]; in fish livers, non-sugar substances can be converted to glucose by gluconeogenesis. At the same time, glucose can also be polymerized into glycogen, stored in the liver. When the body needs, glucose is the main energy material of most cells, through glycolytic process metabolism, to provide energy for the body to maintain normal physiological activities of the body. Therefore, many metabolic studies have found that glucose can often be used as a marker of metabolic changes in the body. For example, Wanger et Al. studied the effects of sesamin on the liver and white muscle metabolism in *Salmo salar* using ^1^H-NMR metabolomics, the metabolism level of metabolic products related to energy metabolism such as glucose, glycogen, and leucine changed significantly [53]. In *Acipenser baerii* exposed to osmotic stress, it was found that the changes of main metabolic products caused by salt acclimation were related to amino acids, osmotic pressure, and energy metabolism [54]. Additionally, different levels of protein in the diet will lead to different activities of key enzymes in glycolysis and changes in blood sugar content in the body. Wang et al. reported that *Pseudosciaena crocea* fed a high protein diet could regulate the activities of key glycolytic and gluconeogenic enzymes, thereby effectively reducing the blood sugar level [55]. In view of the relationship between dietary protein and carbohydrate, there are many studies focusing on dietary diets with protein. Studies on energy in fish nutrition showed that the content of protein in the diet was closely related to the carbohydrate metabolism of fish [56,57,58]. Here, through the analysis of the content of metabolites in the liver, it was found that the only significantly different metabolite was glucose, indicating that the content of protein in the diet could not only reflect the energy metabolism level of tilapia to some extent but also affect the carbohydrate metabolism of tilapia.

In this study, 47 metabolites in tilapia serum were also qualitatively and quantitatively analyzed. It was found that there were significant differences among three metabolites in different groups, namely tyrosine, guanosine, and inosine, which are involved in amino acid metabolism, nucleotide metabolism, and energy metabolism.

As the basic unit of protein, amino acids play a very important role in metabolic activities. Many studies have found that amino acid metabolism is significantly affected by the protein content in the diet and the protein from different sources [59,60]. However, amino acids and their derivatives in 15 liver samples analyzed in this study did not show significant differences among the three treatments, but only tyrosine in serum was found to be significantly different. Many studies have confirmed that the changes in serum tyrosine content were significantly influenced by the activity of proteases. Proteases can affect the stability of tyrosine in animals fed high protein diets [61]. In our study, the reason may be that the protease activity of tilapia increases first and then decreases along with the increase in dietary protein level, which results in the increase in tyrosine content and then decrease. Tyrosine is an aromatic cyclic amino acid, also known as a glucogenic amino acid. The conversion of tyrosine to glucose occurs primarily in the liver. Studies have shown that high protein levels in diets increase the activity of alanine transaminase and aspartate transaminase in the liver. Higher protein levels in diets could improve amino acid metabolism in fish livers [62,63]. Combined with results that the highest levels of glucose in differential liver metabolite was found in the high-protein Group in this study, the difference in Serum tyrosine levels in the three groups may be due to increased liver metabolism in the HP group for improved efficiency of conversion of tyrosine to glucose.

Inosine, also known as hypoxanthine nucleoside, is a purine derivative that functions in the liver and acts as an antioxidant by increasing the superoxide dismutase level in tissues; an elevated inosine level may promote liver function [64,65]. Guanosine, like inosine, belongs to purine nucleosides and is involved in nucleotide metabolism, energy metabolism, protein synthesis, and immune regulation. In multiple metabolisms, guanosine, inosine, and their derivatives affect each other. In our study, the inosine level and guanosine level in the MP group was significantly lower than that of the other groups, combined with the results of the growth of tilapia, suggesting that too low or high levels of this derivative increase metabolic burden and are not conducive to the health of tilapia. They can be used as markers of feed protein metabolism. Nevertheless, further studies are needed to test this hypothesis.

## 5. Conclusions

In summary, dietary protein level had no obvious effect on the diversity of the gut microbiome but had a certain effect on the abundance of different microbial species. Dietary protein can also affect glycolysis and amino acid metabolism in tilapia by altering the level of glucose in the liver, as well as the serum levels of tyrosine, inosine, and guanosine in tilapia.

## Figures and Tables

**Figure 1 animals-11-01024-f001:**
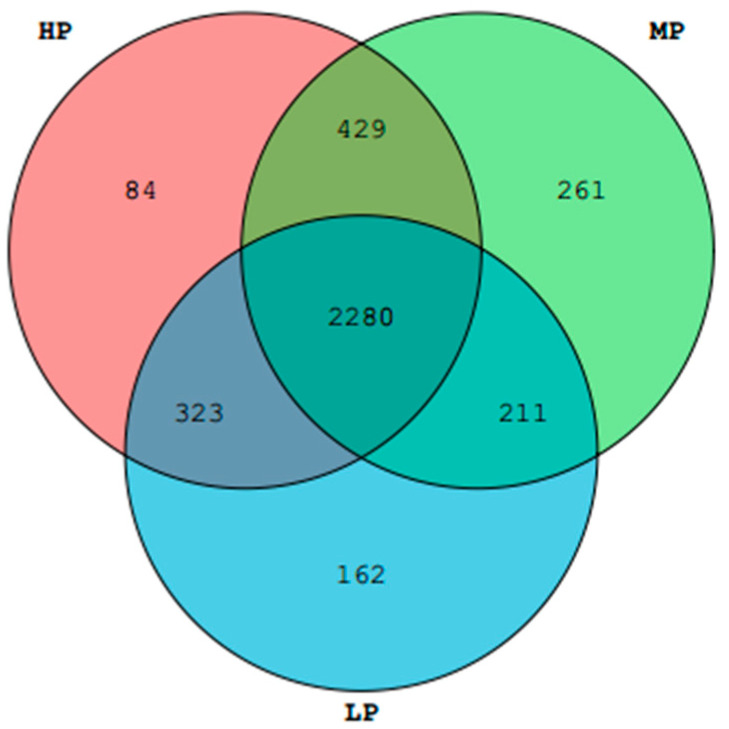
Shared operational taxonomic unit (OTU) analysis of different groups (the low dietary protein group (LP), moderate dietary protein group (MP), high dietary protein group (HP)).

**Figure 2 animals-11-01024-f002:**
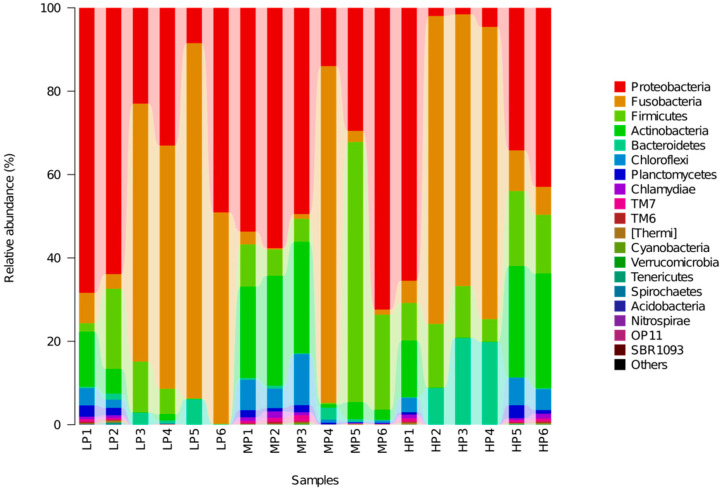
Relative abundance of intestinal bacterial communities at the phylum level.

**Figure 3 animals-11-01024-f003:**
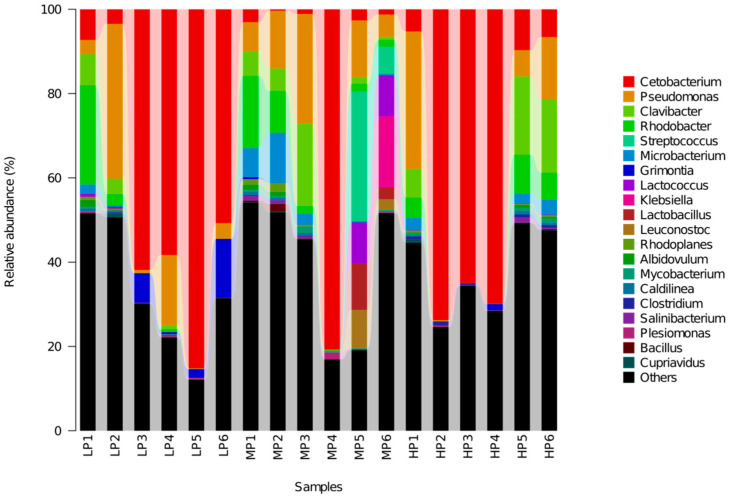
Relative abundance of intestinal bacterial communities at the genus level.

**Figure 4 animals-11-01024-f004:**
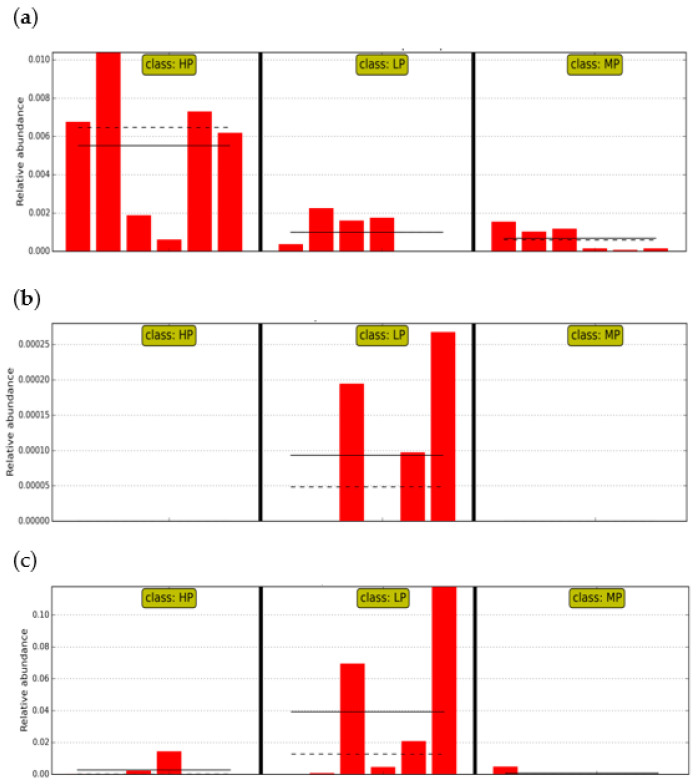
The relative abundance of taxa with significant differences in different groups at the genus level. (**a**) *Clostridium*; (**b**) *Enterovibrio*; (**c**) *Grimontia*.

**Figure 5 animals-11-01024-f005:**
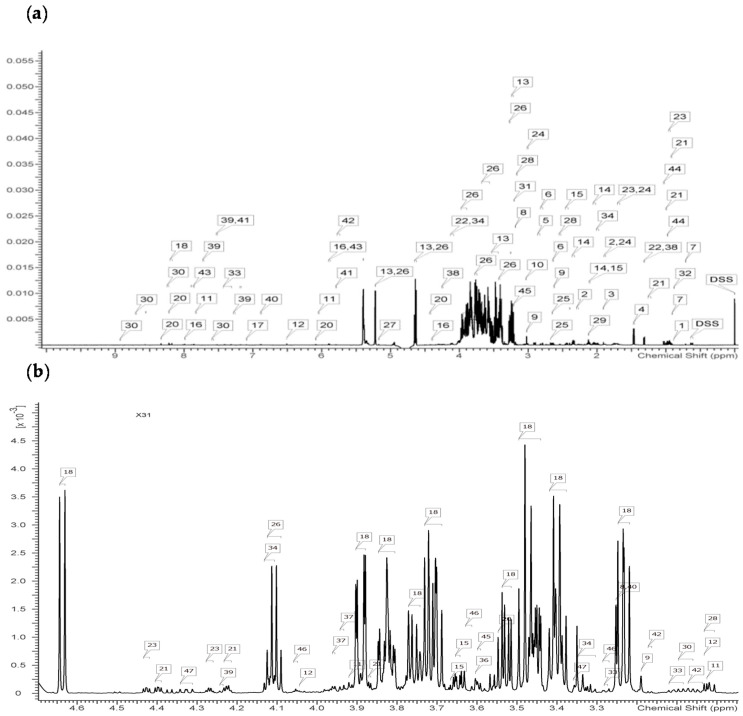
NMR spectrum of tilapia liver and serum. (**a**) NMR spectrum of tilapia liver. 1.2-Hydroxybutyrate; 2: 4-Aminobutyrate; 3: Acetate; 4: Alanine; 5: Asparagine; 6: Aspartate; 7: Cholate; 8: Choline; 9: Creatine; 10: Creatinine; 11: Cytidine; 12: Fumarate; 13: Glucose; 14: Glutamate; 15: Glutamine; 16: Guanosine; 17: Histidine; 18: Hypoxanthine; 19: IMP; 20: Inosine; 21: Isoleucine; 22: Lactate; 23: Leucine; 24: Lysine; 25: Malate; 26: Maltose; 27: Mannose; 28: beta-Alanine; 29: Methionine; 30: Nicotinurate; 31: O-Phosphocholine; 32: Pantothenate; 33: Phenylalanine; 34: Proline; 35: Serine; 36: Succinate; 37: Taurine; 38: Threonine; 39: Tryptophan; 40: Tyrosine; 41: Uracil; 42: Urea; 43: Uridine; 44: Valine; 45: sn-Glycero-3-phosphocholine. (**b**) NMR spectrum of tilapia serum. 1: 1;6-Anhydro-β-D-glucose; 2: 2-Hydroxybutyrate; 3: 2-Hydroxyisobutyrate; 4: 3-Hydroxybutyrate; 5: Acetate; 6: Alanine; 7: Arginine; 8: Betaine; 9: Choline; 10: Citrate; 11: Creatine; 12: Creatinine; 13: Cytidine; 14: Dimethylamine; 15: Ethanol; 16: Formate; 17: Fumarate; 18: Glucose; 19: Glutamine; 20: Glycine; 21: Guanosine; 22: Hypoxanthine; 23: Inosine; 24: Isobutyrate; 25: Isoleucine; 26: Lactate; 27: Leucine; 28: Lysine; 29: Mannose; 30: tau-Methylhistidine; 31: Methionine; 32: Pantothenate; 33: Phenylalanine; 34: Proline; 35: Pyruvate; 36: Sarcosine; 37: Serine; 38: Succinate; 39: Threonine; 40: Trimethylamine N-oxide; 41: Tryptophan; 42: Tyrosine; 43: Uracil; 44: Uridine; 45: Valine; 46: myo-Inositol; 47: trans-4-Hydroxy-L-proline.

**Figure 6 animals-11-01024-f006:**
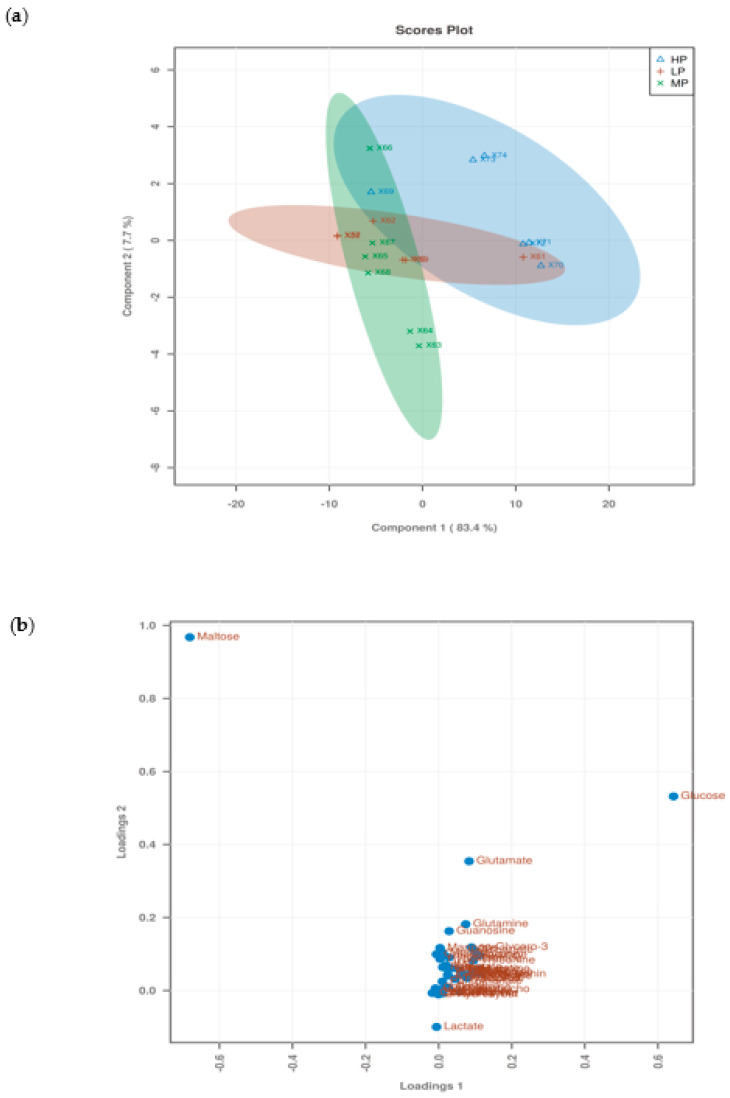
PLS-DA score plots and loading plots of liver data. (**a**) PLS-DA score plots; (**b**) PLS-DA loading plots.

**Figure 7 animals-11-01024-f007:**
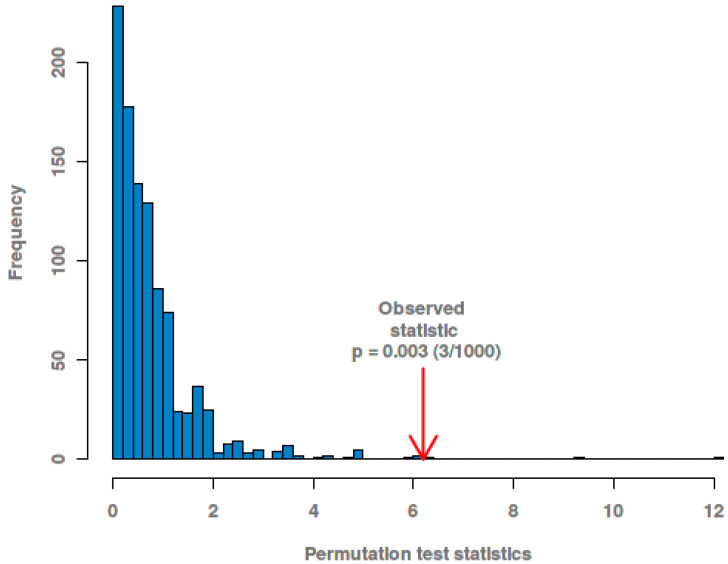
The PLS-DA permutation test of liver data.

**Figure 8 animals-11-01024-f008:**
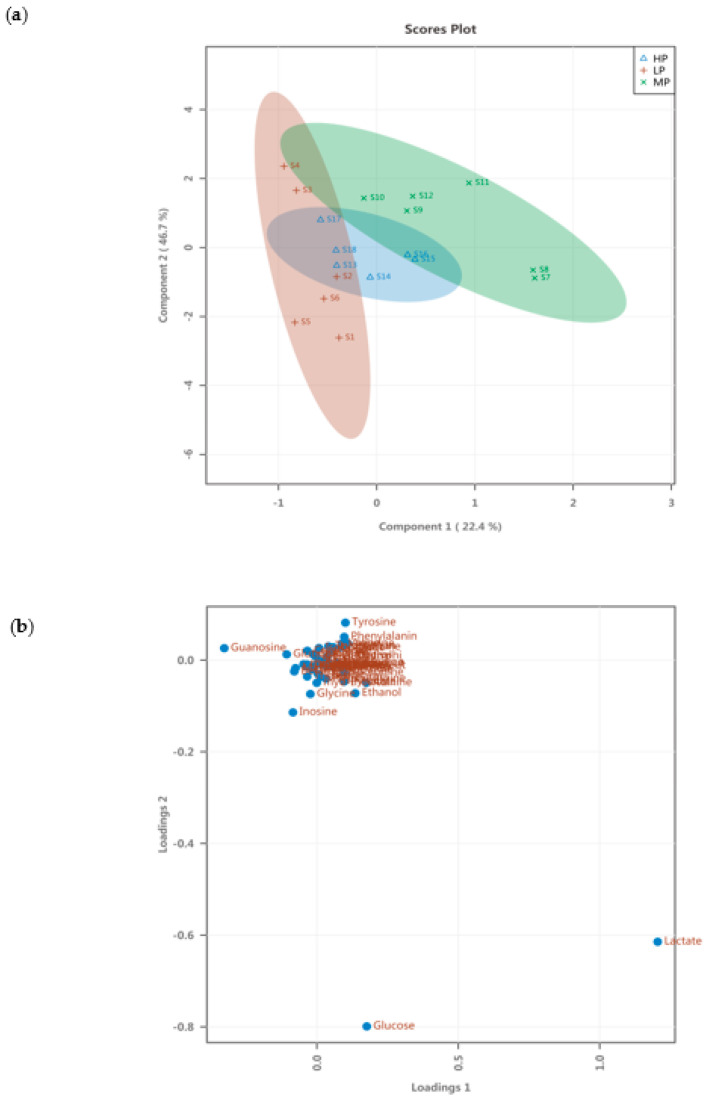
PLS-DA score plots and loading plots of serum. (**a**) PLS-DA score plots; (**b**) PLS-DA loading plots.

**Figure 9 animals-11-01024-f009:**
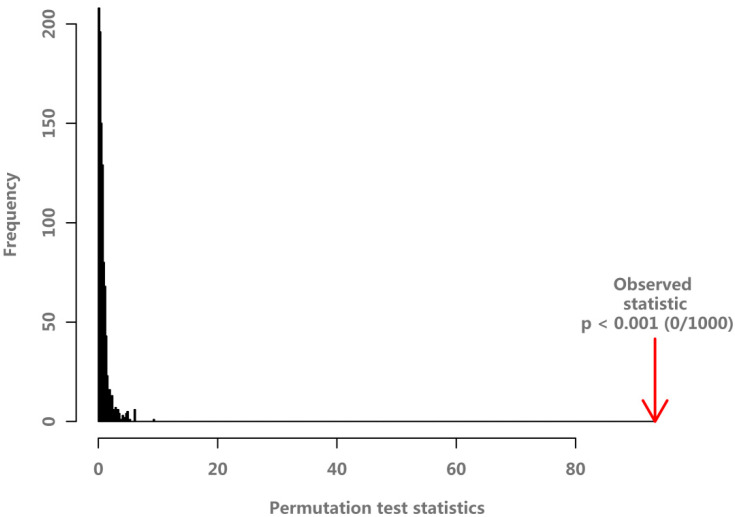
The PLS-DA permutation test for serum data.

**Figure 10 animals-11-01024-f010:**
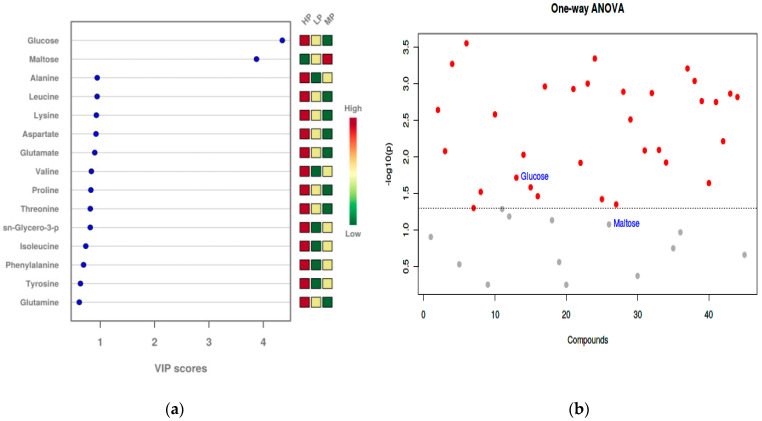
Metabolites variable importance of projection (VIP) Analysis and ANOVA test of metabolites (VIP > 1) in liver. (**a**) Metabolites VIP Analysis; (**b**) one-way ANOVA test of metabolites (VIP > 1). Metabolites above the dotted line had significant differences.

**Figure 11 animals-11-01024-f011:**
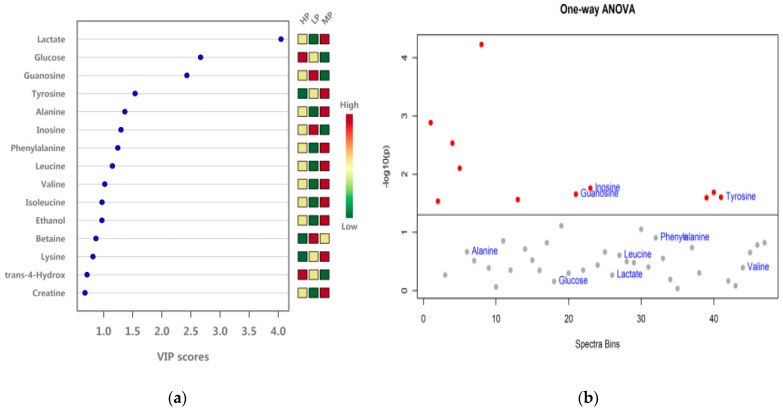
Metabolites VIP Analysis and ANOVA test of metabolites (VIP > 1) in serum. (**a**) Metabolites VIP Analysis; (**b**) one-way ANOVA test of metabolites (VIP > 1). Metabolites above the dotted line had significant differences.

**Table 1 animals-11-01024-t001:** Formulation and proximate composition of the experimental diets.

Ingredient (g/kg)	Dietary Protein Level		
	LP	MP	HP
^a^ Compound protein	250.0	375.0	500.0
^b^ Dextrin	548.6	411.3	274.2
^c^ Fish oil	45.8	43.8	41.7
^d^ Soybean oil	50.0	50.0	50.0
^e^ Vitamin premix	10.0	10.0	10.0
^f^ Mineral premix	10.0	10.0	10.0
^g^ Monocalcium phosphate	20.0	20.0	20.0
^g^ Choline chloride	1.0	1.0	1.0
^g^ Micro-cellulose	60.6	74.9	89.1
^g^ Titanium dioxide	4.0	4.0	4.0
Total (g)	1000	1000	1000
Proximate composition (g/kg diet as fed)			
Crude protein	209.6	302.4	401.9
Crude lipid	98.0	96.9	99.3
Moisture	77.8	85.1	81.0
Ash	31.0	30.5	40.2
Gross energy (MJ/kg)	19.4	20.2	20.1

^a^ Compound protein contained casein (Xilong Chemical Co., Shantou, China), fish meal (Wuhan Coland), and gelatin (Xilong Chemical Co.) at a ratio of 4:1:1. ^b^ Xilong Chemical Co. ^c^ Wuhan Coland Co; ^d^ COFCo. ^e^ The vitamin mixture supplied the following (mg/g mixture): thiamine hydrochloride, 5; riboflavin, 5; calcium pantothenate, 10; nicotinic acid, 6.05; biotin, 0.03; pyridoxine, 4; folic acid, 1.5; inositol, 200; L-ascorbyl-2-polyphosphate, 3.95; tocopherol, 5; menadione, 4; retinol, 0.4; cholecalciferol, 18.74 IU/g. All ingredients were diluted with micro-cellulose to 1 g. ^f^ The mineral mixture supplied the following (mg/g mixture): C_6_H_10_CaO_6_, 500; FeSO_4_·7H_2_O, 20; MgSO_4_, 100; NaH_2_PO_4_, 100; NaCl, 20; AlCl_3_, 0.6; KIO_3_, 0.6; KCl, 40; CuSO_4_, 2; MnSO_4_, 4; CoCl_2_, 2; ZnSO_4_, 20. All ingredients were diluted with micro-cellulose to 1 g. ^g^ China national pharmaceutical group corporation.

**Table 2 animals-11-01024-t002:** Parameters of NMR.

Item	Value
Temperature (K):	298.01
Magnet Frequency (MHz):	600.2
Transients/Scans:	128
Frequency Domain Size:	131,072
Spectral Width:	8403.361
Time Domain Size:	65,536
Pulse Sequence:	noesygppr1d

**Table 3 animals-11-01024-t003:** Growth Performance of tilapia fed different test diets.

Item	LP	MP	HP
IBM/g	39.70 ± 0.54	38.87 ± 0.47	39.52 ± 0.32
FBM/g	178.21 ± 4.36 ^a^	194.13 ± 1.28 ^b^	200.59 ± 8.63 ^b^
WGR/%	348.84 ± 5.08 ^a^	399.54 ± 7.71 ^b^	407.57 ± 19.56 ^b^
SGR/%▪d−1	1.85 ± 0.02 ^a^	1.99 ± 0.02 ^b^	2.00 ± 0.05 ^b^

Note: Means in the same row sharing the same or none superscript letter are not significantly different, as determined by Tukey’s test (*p* > 0.05). WGR: weight gain rate; FBM: final body weight; IBM: initial body weight; SGR: specific growth rate.

**Table 4 animals-11-01024-t004:** Sequencing number per sample.

Group	Sample	Sequence Number
LP(the low dietary protein group)	LP1	52,619
	LP2	52,611
	LP3	54,177
	LP4	56,198
	LP5	61,781
	LP6	66,691
MP(the moderate dietary protein group)	MP1	69,195
	MP2	68,488
	MP3	62,518
	MP4	66,462
	MP5	62,041
	MP6	55,079
HP(the high dietary protein group)	HP1	55,529
	HP2	53,697
	HP3	62,154
	HP4	68,581
	HP5	60,767
	HP6	71,516
	total	1,100,104

**Table 5 animals-11-01024-t005:** Alpha diversity analysis of tilapia intestinal microbial sample sequencing data.

Samples	Simpson	Chao1	ACE	Shannon
LP	0.9526 ± 0.0157	1146.27 ± 182.16	1049.83 ± 183.28	6.3617 ± 0.64
MP	0.9540 ± 0.0240	1305.95 ± 391.41	1317.52 ± 382.60	6.5783 ± 0.84
HP	0.9469 ± 0.0176	1314.75 ± 408.66	1313.77 ± 425.02	6.3583 ± 0.74

## Data Availability

All data used in the current study are available from the corresponding author on reasonable request.

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
