# Peer review of "Effects of Dietary Protein Level on the Gut Microbiome and Nutrient Metabolism in Tilapia (Oreochromis niloticus)"

_animals, 2021, doi:10.3390/ani11041024_

Round 1

Reviewer 1 Report

Grammar & Spelling

L16: "... dietary with different..."

L21: "... a low(30%), moderate (20%),..."

L44: "And it was a main farmed fishes in China"

L55/56: "... have all been used to the auqatic nutrition research..."

L56: "... there had some further..."

L187 & 188: please change to "indices"; not "indexes"

Overall comments

L53-64: What are the reasons for this paragraph? It does not provide useful information for the research question.

The aims of this study are stated in the introduction, however it remains unclear what is the overall purpose of the study?

L109: What are "intestinal samples"? Biopsy? Fecal samples? The type of sample is crucial for microbiome analysis.

L143-160: The section about the statistical analysis is much too short and misses explanations/justifications for many of the used methods as well as descriptions of exact bioinformatic/statistical tools.

L161-164: Please remove.

L177: no of sequences varied from... between what?

L183: please rephrase table heading; please rephrase "sequencing quantity"

L203: significant differences among which genera? and where is this shown?

Figures 5 & 7: too small and low resolution

The discussion section does not adequately adress the microbiome results in relation to other studies (only one is basically mentioned in more detail). Also the results are not discussed in relation to the aims of the study or what the findings actually mean for tilapia aquaculture. This needs significant improvement. Furthermore, the outcome of the NMR analysis is not sufficiently discussed.

Reviewer 2 Report

The manuscript is in general well written but improvements are needed in some points. The methodologies used are adequate but must be further described. The results must be better explored and discussed to support the main conclusions.

For example, the authors state in their abstract that they found differences in the microbial composition of the gut....but since those differences were not statistically significant, such sentence can be farfetched and should be rephased. ASuthors should be conservative in their assumptions.

Further details on the NGS data analysis (e.g .  taxonomic database on which the results were based) should be given. Did the authors considered removing singletons and low abundant OTUs? Did the authors consider including ASVs as well?

An Accession No. for the 16S rRNA raw reads deposit in a database such as SRA (Sequence Read Archive)should be given.

Line 21: % of diet with low and moderate are swapped, please correct

Lines 162-164 (This section may be divided by subheadings. It should provide a concise and precise 162 description of the experimental results, their interpretation, as well as the experimental 163 conclusions that can be drawn. “)  should be removed.

Table 4: The word “statistical” should be removed from the title, since the results included in the table are not from statistical analysis

Line 203: “and there were significant differences among them”: significance of differences must be supported by a statistical test and corresponding p-value. Please add.

Figure 6 and Figure 8 seem exactly the same. Please verify if correct figures were uploaded.

Line 314-316: Please rephrase, it is confusing: “Although in our research, the intestinal microbial diversity had no difference, likely to many other studiesat phyla and genera levels, it was found the proportion of the same microbial in different groups of tilapia intestinal microflora composition is slightlydifferent in our research”.

Line 309: In the discussion  the authors try to explain the absence of differences at the gut microbiota level, with the crude lipid in the diet but give just on example of another study for their comparison and go no further in their interpretation of results. Further references and tentative explanations would enrich the discussion. Also in Line 312 the authors try to explain it in the light of the differences in initial fish-weigh. Again further development, with appropriate references would enrich discussion.

Line 318:please explain which microorganism taxa is called Pachyderm and provide the reference that states it to be prevalent in.fish gut communities.  

Line 3234: “ and so on” ? please explain

Line 328: please elaborate “and some species can also lead to some diseases”. A scientific accurate language would be preferable

There are some scientific imprecision calling glucose a carbohydrate, or carbohydrate a metabolite ( examples on Lines 354 “The difference metabolite was found to be carbohydrate, which indicated that the protein content in the  diet had a significant effect on the carbohydrate metabolism of tilapia liver.” or Line 378 “the significant different metabolite metabolite was carbohydrate”. It is inaccurate, please correct it.

Guanosine an Iosine differences could be further explored, in the same proportion as tyrosine to turn the discussion more interesting and informative.

Round 2

Reviewer 1 Report

The Simple Summary and the Abstract state, that "... there were differences in the microbial composition of the gut.". This is misleading, as there were no significant differences between microbial compositions observed (and are thus not different). Only three individual bacterial genera were differently abundant between diatary groups. However, statistical analysis is difficult with a sample size of two individuals per tank (Figures 2 & 3 indicate "pairing" of samples, which might be due to tank effects!) and should therefore be considered in the discussion of results.

2.2 Sample collection:
a) The fish were fasted for 24 hrs. This is crucial, as fasting can have a high impact on the intestinal microbiome! This should be considered when discussing the results.

b) Midgut samples without contents were collected. If I understand correctly, then only gut tissue samples without faeces were collected. Please explain why this way of sample collection was chosen and not a combination of tissue (better: mucus only) and faeces?

2.5 Statistical analysis
a) Why were rarefaction curves generated and where are they shown? Which R package was used for this analysis?

b) Which program/package was used for the LEfSe analysis?

L197: The BioProject ID is inaccessible.

Figures 6 & 8: text is still too small and cannot be read

L372-376: The example of some disease-causing Fusobacteria species is no explanation for reduced proportions of Fusobacteria in the MP dietary group. Please reconsider.

L377-384: This paragraph is highly speculative and no change in the Proteobacteria abundance has been observed! Thus, there is no imbalance of the intestinal flora. Please reconsider.

5. Conclusions:
"... dietary protein can promote [...] the breakdown of nutrients by microorganisms." -> this has not at all been examined in this study and is therefore miesleading to readers. Please exclude the entire sentence and rewrite the conclusions according to the actual results.

Please check for further text editing:

L27-29

L66-68

L124

L178

L200 -> please extend Figure Capture (what groups?...etc.)

L205 -> please extend Table Heading (what is shown?)

L221 (what proportions?)

L342-344

L361-363

Author Response

Comments and Suggestions for Authors

The Simple Summary and the Abstract state, that "... there were differences in the microbial composition of the gut.". This is misleading, as there were no significant differences between microbial compositions observed (and are thus not different). Only three individual bacterial genera were differently abundant between diatary groups. However, statistical analysis is difficult with a sample size of two individuals per tank (Figures 2 & 3 indicate "pairing" of samples, which might be due to tank effects!) and should therefore be considered in the discussion of results.

Answer: In our experiment, six individuals from three tanks (two individuals per tank) in each group were random selected. It was a comparative analysis of 18 fish in 3 groups(six individuals per group). Therefore, we think the results have certain representativeness.

2.2 Sample collection:
a) The fish were fasted for 24 hrs. This is crucial, as fasting can have a high impact on the intestinal microbiome! This should be considered when discussing the results.

Answer: We explained it in our discussion.

  1. b) Midgut samples without contents were collected. If I understand correctly, then only gut tissue samples without faeces were collected. Please explain why this way of sample collection was chosen and not a combination of tissue (better: mucus only) and faeces?

Answer: In our description for samples, it may not be accurate enough. The fact is that we only collected the midgut and did nothing else on it after fasted for 24 h, for example, to further remove the content completely. The reason for this sampling procedure is that we don’t want other operation to disturb the gut microbes experimental data, and also eliminate the interference of food. And we had amended.

2.5 Statistical analysis
a) Why were rarefaction curves generated and where are they shown? Which R package was used for this analysis?

Answer: In our research, rarefaction curve analysis was used to evaluate whether the sequencing depth is enough to cover the entire microbiome community. However, due to the limitation of the length of the article, the result is not shown. In method, we have deleted this sentence.

  1. b) Which program/package was used for the LEfSe analysis?

Answer: We had amended.

L197: The BioProject ID is inaccessible.

Answer: The information for the BIOPROJECT ID PRJNA700685 has been released and is now available.

Figures 6 & 8: text is still too small and cannot be read

Answer: We had amended.

L372-376: The example of some disease-causing Fusobacteria species is no explanation for reduced proportions of Fusobacteria in the MP dietary group. Please reconsider.

Answer: We had amended.

L377-384: This paragraph is highly speculative and no change in the Proteobacteria abundance has been observed! Thus, there is no imbalance of the intestinal flora. Please reconsider.

Answer: We had amended.

  1. Conclusions:
    "... dietary protein can promote [...] the breakdown of nutrients by microorganisms." -> this has not at all been examined in this study and is therefore miesleading to readers. Please exclude the entire sentence and rewrite the conclusions according to the actual results.

Answer: We had amended.

Please check for further text editing:

L27-29

L66-68

L124

L178

L200 -> please extend Figure Capture (what groups?...etc.)

L205 -> please extend Table Heading (what is shown?)

L221 (what proportions?)

L342-344

L361-363

Answer: We had amended.

Reviewer 2 Report

The authors have includes the suggestions and corrections made by the reviewer with exception of the  accession number for data deposit at NCBI-SRA provided (BioProject ID PRJNA700685)that does not retrieve any result when checked at the NCBI website. Authors should provide a valid accession number before acceptance for publication. 

Author Response

Comments and Suggestions for Authors

The authors have includes the suggestions and corrections made by the reviewer with exception of the  accession number for data deposit at NCBI-SRA provided (BioProject ID PRJNA700685)that does not retrieve any result when checked at the NCBI website. Authors should provide a valid accession number before acceptance for publication. 

Answer: The information for the BIOPROJECT ID PRJNA700685 has been released and is now available.